# Improved Device Distribution in High-Performance SiN_x_ Resistive Random Access Memory via Arsenic Ion Implantation

**DOI:** 10.3390/nano11061401

**Published:** 2021-05-25

**Authors:** Te-Jui Yen, Albert Chin, Vladimir Gritsenko

**Affiliations:** 1Department of Electronics Engineering, National Yang Ming Chiao Tung University, Hsinchu 300, Taiwan; yenrick42269.ee05g@g2.nctu.edu.tw; 2Rzhanov Institute of Semiconductor Physics, Siberian Branch, Russian Academy of Sciences, 630090 Novosibirsk, Russia; grits@isp.nsc.ru; 3Novosibirsk State University, 2 Pirogov Street, 630090 Novosibirsk, Russia; 4Novosibirsk State Technical University, 20 Marks Avenue, 630073 Novosibirsk, Russia

**Keywords:** SiN_x_ RRAM, ion implantation, neuron mimicking device

## Abstract

Large device variation is a fundamental challenge for resistive random access memory (RRAM) array circuit. Improved device-to-device distributions of set and reset voltages in a SiN_x_ RRAM device is realized via arsenic ion (As^+^) implantation. Besides, the As^+^-implanted SiN_x_ RRAM device exhibits much tighter cycle-to-cycle distribution than the nonimplanted device. The As^+^-implanted SiN_x_ device further exhibits excellent performance, which shows high stability and a large 1.73 × 10^3^ resistance window at 85 °C retention for 10^4^ s, and a large 10^3^ resistance window after 10^5^ cycles of the pulsed endurance test. The current–voltage characteristics of high- and low-resistance states were both analyzed as space-charge-limited conduction mechanism. From the simulated defect distribution in the SiN_x_ layer, a microscopic model was established, and the formation and rupture of defect-conductive paths were proposed for the resistance switching behavior. Therefore, the reason for such high device performance can be attributed to the sufficient defects created by As^+^ implantation that leads to low forming and operation power.

## 1. Introduction

In the past few decades, resistive random access memory (RRAM) [1,2,3,4,5,6,7,8,9,10,11,12,13,14,15,16,17,18,19,20,21,22,23,24,25,26] has attracted massive attention due to its simple process, high density, multilevel state, high operation speed, and low power consumption [1,2,3,4,5]. The RRAM device has high potential to be implemented for artificial intelligence (Al) and neuromorphic computing [6,7] in several kinds of emerging memories. Resistance switching behaviors are highly related to the materials of switching layer and electrodes. Numerous materials can serve as resistance switching layers, such as AlO_x_ [8], HfO_x_ [9], GeO_x_ [10,11], TiO_2_ [12], SiO_x_ [13], and SiN_x_ [14,15,16]. However, the resistance state is distributed randomly and difficult to control. Such a large resistance distribution limits the vitally important memory array circuit size [17,18]. In order to overcome this challenge, we pioneered the GeO_x_ RRAM device, which performed relatively improved distribution [10,11,12]. In addition, the all-non-metal SiN_x_ RRAM device was demonstrated, which reached reasonable distribution [15]. SiN_x_ was chosen as a switching layer due to its wide usage in integrated circuit as the passivation layer and charge storage layer in NAND flash memory. However, previous SiN_x_ works [14,15,16], like other dielectric RRAM devices, did not exhibit good distribution for memory array application [17,18]. Here we introduce a novel method to address this issue. It is important to notice that high compliance current (I_cc_) and high forming voltage will damage the dielectric layer and create unrecoverable defects, which will lead to poor retention time, decreased endurance cycles, and wide resistance distribution. Based on the discussion above, we improved the RRAM device’s integrity by implanting arsenic (As) ions into the SiN_x_ layer. The reason for choosing As ion (As^+^) for implantation is its heavy atomic mass to create defects and a short stopping range within the switching layer. The As^+^ implantation into SiN_x_ cannot be used as a dopant, which is quite different from the As^+^ implantation for the Si metal–oxide–semiconductor field-effect transistor. Besides, As is a semimetal; however, its property becomes that of a semiconductor with a band gap of 1.2–1.4 eV if amorphized [27], which is used for ion implantation in this case. As^+^-implanted RRAM devices exhibited high stability of pulsed endurance and excellent 85 °C retention. In addition, tight set and reset voltage (V_set_ and V_reset_) distributions were achieved, which is crucial for the application of a large-size cross-point memory array circuit [17,18].

## 2. Materials and Methods

A highly p-type doped P^+^ silicon wafer with a resistivity of 0.001–0.005 ohm-cm was prepared and used as a bottom electrode. Through the standard RCA clean process, the native oxide on the P^+^ silicon wafer was removed. Subsequently, a 35 nm thick SiN_x_ was deposited via plasma-enhanced chemical vapor deposition (PECVD) under a 300 °C temperature, with a NH_3_, SiH_4_ (8% in Ar), and N_2_ gas flow of 6, 125, and 200 sccm, respectively. Then, the As ions were implanted into a SiN_x_ layer at a 10 keV energy and 10^15^ cm^−2^ dose. Finally, a 50 nm Ni layer was deposited through an electron beam evaporator and used as the top electrode. The ion implantation doping concentration and energy can influence the number of created defects and the penetration depth of the As ions, respectively. The 35 nm SiN_x_ layer was chosen from the stopping and range of ions in matter (SRIM) simulation to avoid As^+^ penetration into the bottom electrode at an implantation energy of 10 keV. This energy value is the lowest energy available for a typical ion implantation equipment. An ion implantation doping concentration of 5 × 10^15^ cm^−2^ is the highest available concentration in a typical ion implantation equipment. The current–voltage (*I–V*) characteristics were extracted by a semiconductor parameter analyzer (HP4155B) and a probe station. Bias voltage was applied to the top electrode, and the bottom electrode was grounded. AC endurance was obtained by a pulse generator (Agilent 81110A). The implanted As atom profile and the created defect profile inside the SiN_x_ layer were obtained from the SRIM simulation. Appendix A displays top and cross-sectional view images from scanning electron microscopy (SEM) and transmission electron microscopy (TEM), respectively. A device dimension of 120 μm diameter was obtained, and a SiN_x_ layer thickness of 35 nm was measured.

## 3. Results

In Figure 1a, the forming process of the As^+^-implanted and normal nonimplanted SiN_x_ RRAM devices is displayed. Before the forming process in Figure 1a, the resistances of the As^+^-implanted and nonimplanted SiN_x_ RRAM devices were 3.45 × 10^9^ and 37.5 × 10^9^ ohm at 1 V, respectively. Thus, the resistance of the nonimplanted device was 10.9 times higher than that of the As^+^-implanted device before forming. During the forming process, a relatively high voltage was applied to break the Si–N bonds and created the defect-conductive path for electrons. The forming current and voltage were highly related to device performance. In the forming process of the nonimplanted device, I_cc_ was increased from 100 μA with a 100 μA increment. When I_cc_ increased to 500 μA, the resistance state was able to switch to LRS. In sharp contrast, the resistance state of the As^+^-implanted device can be switched to LRS even at a low 100 μA. The forming voltage of the As^+^-implanted RRAM device was 7.1 V under 100 μA I_cc_. In contrast, the nonimplanted SiN_x_ device needed a high 500 μA I_cc_ current and 11 V applied voltage to achieve the forming process. Thus, the As^+^-implanted SiN_x_ device exhibited a significantly lower initial power than the nonimplanted devices, which was due to a large number of defects created by ion implantation for current conduction. Such low forming power will decrease the nonrecoverable defects, which is important for device switching performance. A negative voltage applied to the top electrode causes a slightly higher forming voltage and is unfavorable for low power operation. Appendix A shows the device-to-device distribution of the forming voltage. The mean value (μ) of the As^+^-implanted and nonimplanted devices are 7.03 and 10.96 V, respectively. Besides, the standard deviation (σ) is largely improved by As^+^ implantation. After the forming process, the reset and set processes were executed to switch the resistance states of the RRAM devices. The higher high- and low-resistance-state (HRS and LRS) currents of the nonimplanted RRAM device, after the forming process, resulted from the excessive defects created by the high forming voltage and current. Figure 1b displays the set and reset characteristics of the As^+^-implanted and normal SiN_x_ RRAM devices. The RRAM device implanted with As ions exhibited a lower reset current and voltage than the nonimplanted device. Such lower reset power in combination with low forming power can decrease the unrecoverable damage to the switching layer during the device set–reset operation. On the other hand, the nonimplanted SiN_x_ device shows higher HRS and LRS currents, which are related to excess electric-field-induced defects from the high forming voltage and high I_cc_. The high defects not only increase the off-state HRS leakage current, but also increase both the voltage and current to switch the LRS back to HRS. Such high HRS and LRS currents will decrease the on-state/off-state resistance window, which is crucial for high-memory-density multiple-state memory array.

Figure 2a,b shows the 85 °C retention and pulsed endurance characteristics, respectively, which are the crucial performance indexes for RRAM devices. For retention measurements, the device was switched to LRS, and the read voltage was applied at 1, 10, 100, 1000 and 10,000 s to record the LRS current. The same methodology was applied to measure the HRS current. The on-state/off-state resistance window of the As^+^-implanted device showed a slight decrease from 1.96 × 10^3^ to 1.73 × 10^3^ after a 10^4^ s retention test at 85 °C. In sharp contrast, the normal SiN_x_ RRAM device showed severe degradation, which decreased from 3.95 × 10^2^ to 45 after 10^4^ s retention at 85 °C. To switch the resistance state properly during the endurance test, pulse voltages of +6 and −6 V with a 2 μs width were applied for both devices. For the pulsed cycling test, the resistance window of normal SiN_x_ RRAM devices was decreased rapidly from 1.4 × 10^2^ to 17 after a 10^4^ cycle operation. For comparison, the As^+^-implanted SiN_x_ RRAM device exhibited an excellent 10^3^ resistance window, even after 10^5^ endurance cycles. The poor retention and endurance data of the normal SiN_x_ RRAM device can be attributed to high reset current and high forming voltage.

In Figure 3a,b, we analyzed the device-to-device and cycle-to-cycle V_set_–V_reset_ distributions of the As^+^-implanted and nonimplanted SiN_x_ RRAM devices. The coefficient of variation (CV) was used to evaluate the distribution, which was defined as the σ divided by μ (CV = σ/|μ| × 100%). For the RRAM devices, lower CV values of device-to-device and cycle-to-cycle result in better uniformity and stability, respectively. The operation voltages of the device-to-device distribution were obtained from the average of the first 10 cycles of the 25 devices. The V_set_ and V_reset_ CV values of the nonimplanted SiN_x_ RRAM devices were 17% and 19.8%, respectively. On the other hand, the As^+^-implanted device showed better uniformity with 10.7% and 9.8% V_set_ and V_reset_ CV values. For the cycle-to-cycle measurements, a voltage sweep rate of 0.5 V/s was used. For the As^+^-implanted device, the V_set_ and V_reset_ were ramped at 0~4 and 0~−2 V, respectively. The V_set_ and V_reset_ of the nonimplanted device were ramped at 0~5 and 0~−3 V, respectively. For the cycle-to-cycle distribution, the As^+^-implanted device also exhibited excellent V_set_ and V_reset_ CV values of only 2.2% and 3.8%. In sharp contrast, the normal SiN_x_ device exhibited V_set_ and V_reset_ CV values of 8.3% and 7.6% cycle-to-cycle distribution. Appendix A exhibits the cycle-to-cycle and device-to-device distributions of the set–reset resistances (R_set_–R_reset_), respectively. The As^+^-implanted RRAM exhibited significantly tighter distribution of R_set_ and R_reset_ than those of the nonimplanted case. In Table 1, we summarize the SiN_x_ RRAM device distribution data [15,16]. The As^+^-implanted SiN_x_ RRAM device in this work exhibited good device-to-device (D2D) uniformity and excellent cycle-to-cycle (C2C) reliability, which is crucial for memory array circuit and neuron mimicking applications.

To understand the conduction behavior of the As^+^-implanted RRAM devices, the SRIM simulation was performed to calculate the As atom distribution and created defect distribution. The SRIM simulation only allows a maximum of 10^6^ cm^−2^ dose in the simulation owing to the required large computing resource. As shown in Figure 4a, the peak of As atom concentration was centered at 16.7 nm from the SiN_x_ surface and decreased to negligible from 16.7 to 35 nm. The 35 nm SiN_x_ layer was chosen from the SRIM simulation to avoid the As^+^ penetration into the bottom electrode at an implantation energy of 10 keV. This energy value is the lowest energy available for a typical ion implantation equipment. It is important to notice that the implanted As ions will break massive numbers of the Si–N bonds and create defects at the same time. This is why the created defect profile shown in Figure 4b is similar to the As atom profile in Figure 4a. Consequently, the As^+^-implanted RRAM created more defects within the SiN_x_ layer than the nonimplanted device.

To investigate the electron transport mechanism, the *I–V* curves of the As^+^-implanted and normal SiN_x_ RRAM devices were analyzed. Here, various defect-related conduction mechanisms were fitted. However, some of the fitting results were against physical principles, such as unreasonable dielectric constant and hopping distance, which were observed in the Poole–Frenkel (P–F) emission [20,21] and hopping conduction [22,23,28] mechanisms, respectively. As depicted in Figure 5a, the HRS and LRS currents of the As^+^-implanted SiN_x_ RRAM devices were fitted well with space-charge-limited conduction (SCLC) [23,24,25,26], which can be divided into three regions corresponding to the slopes of 1, 2, and >2. When the slope equals 1 (J ∝ V), the curves follow Ohm’s law, which is expressed as [29]:(1)$$\;J=\frac{qn_{0}\mu V}{d}\;,$$
where $n_{0}$, μ, and d are the free carrier density, electron mobility, and dielectric thickness, respectively. In the low-field region (region I), the current is dominated by free carriers. After the applied voltage was higher than the transition voltage (V_tr_) and lower than the trap-filled limit voltage (V_TFL_), the curve was fitted to slope = 2 (J ∝ V^2^) and expressed as [15,29]: (2)$$J=\frac{9\mu \varepsilon \theta V^{2}}{8d^{3}}\;,$$
where ε and θ are the static dielectric constant and the ratio of the free carrier density to all carrier density. In region II, the free carrier density will increase along with the increasing applied voltage and contributes to the trap-filled limit current. Once all the traps are filled (V > V_TFL_), the current will increase rapidly, which is the trap-free current corresponding to a slope higher than 2 (region III). The HRS and LRS of the nonimplanted device exhibited SCLC and ohmic behavior, respectively, as depicted in Figure 5b. Note that if the Ni electrode plays a role in conduction, the self-rectifying phenomenon would be observed. However, there is no obvious self-rectifying behavior in the measured data. Thus, the formation and rupture of the conducting path are dominated by the defects in the SiN_x_ layer.

The measured *I–V* curves of the As^+^-implanted and nonimplanted devices, before the forming process, were analyzed in Appendix A, respectively. It can be observed that the As^+^-implanted RRAM device exhibited the same SCLC conduction mechanism before and after the forming process. On the other hand, the conduction mechanisms of the nonimplanted SiN_x_ RRAM device before forming is the hopping conduction, which changes to SCLC after the forming process. This is due to a massive number of the defects created by the high forming power.

According to the measured data and the simulation results, the potential microscopic conduction schematic diagram can be constructed as displayed in Figure 6. In the as-fabricated step, the As^+^-implanted SiN_x_ RRAM device shows extra defects created by ion implantation, which corresponds to the measured higher initial state current as depicted in Figure 1a. In addition, the distribution of implant-induced defects in Figure 6a was constructed according to the simulation results shown in Figure 4b. Before the forming step, the nonimplanted device exhibits lower current than the As^+^-implanted device, as shown by a smaller number of as-fabricated defects in Figure 6b. After applying a high forming voltage and a high I_cc_, a massive number of defects were induced randomly for current conduction by the high electric field in the nonimplanted SiN_x_ RRAM device (Figure 6d). In contrast, a relatively low voltage and current were needed to induce sufficient defects to form the current conductive path in the As^+^-implanted SiN_x_ device. From the SRIM simulation results shown in Figure 4b, the As^+^-implantation-induced defects follow a Gaussian distribution in the SiN_x_ layer. The defects in the tail of the Gaussian profile is too low to form a conduction pass. After the set process, extra defects will be formed in the tail region of the Gaussian profile. The current tends to flow via the lowest resistance and through those electric-field-created defects near the bottom of the SiN_x_ layer, the green-dash square region, as depicted in Figure 6c. After the set process, the resistance states were switched from HRS to LRS. Since both the As^+^-implanted and normal SiN_x_ RRAM devices exhibited typical bipolar switching characteristics [19], the negative voltage bias was required. In Figure 1b, the measured current of the As^+^-implanted device decreases rapidly when the reset voltage is close to −1.5 V, representing that the conducting path was ruptured as depicted by the red-dash square region in Figure 6e. Figure 6f shows the HRS case of the nonimplanted SiN_x_ RRAM device. The conducting filament should be dissolved after reset. However, the measured high HRS current in Figure 1b indicates that the conducting filaments were not completely dissolved, resulting in the high leakage current.

## 4. Conclusions

In this work, we compared the SiN_x_ RRAM device with As^+^ implantation with the nonimplanted device. By applying As^+^ implantation, the uniformity and reliability of the SiN_x_ RRAM device can be improved significantly. The 85 °C retention and pulsed endurance tests also exhibited excellent stability. Such high-performance, tight-operation-voltage-distribution, and CMOS-compatible RRAM devices have high potential for memory array circuit and future neuromorphic computing applications.

## Figures and Tables

**Figure 1 nanomaterials-11-01401-f001:**
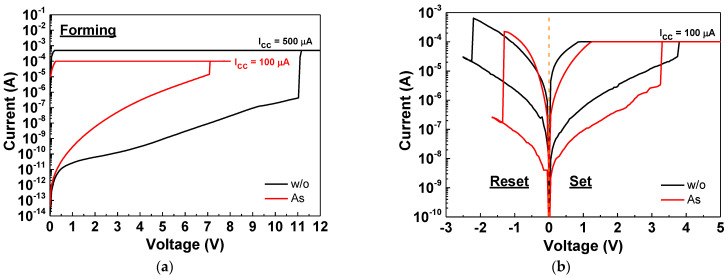
The *I–V* characteristics at (**a**) the forming process and (**b**) the set–reset process of the As^+^-implanted and nonimplanted SiN_x_ RRAM devices.

**Figure 2 nanomaterials-11-01401-f002:**
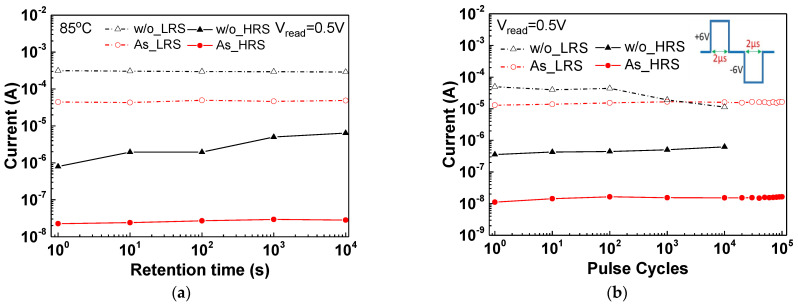
The (**a**) 85 °C retention and (**b**) pulsed endurance characteristics of the As^+^-implanted and nonimplanted SiN_x_ RRAM devices.

**Figure 3 nanomaterials-11-01401-f003:**
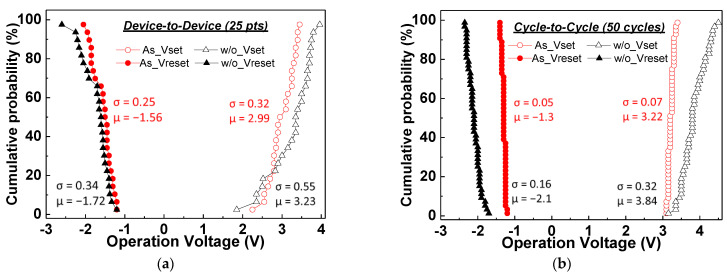
The (**a**) device-to-device and (**b**) cycle-to-cycle of V_set_–V_reset_ distribution of As^+^- and without implanted SiN_x_ RRAM devices.

**Figure 4 nanomaterials-11-01401-f004:**
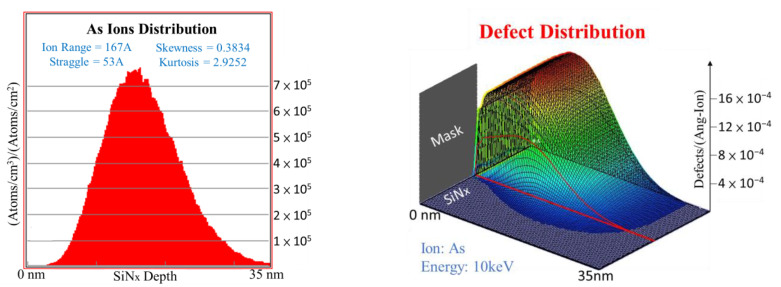
The simulation results of the (**a**) As atoms and (**b**) 3D defect distribution of the As^+^-implanted SiN_x_ layer with a thickness of 35 nm.

**Figure 5 nanomaterials-11-01401-f005:**
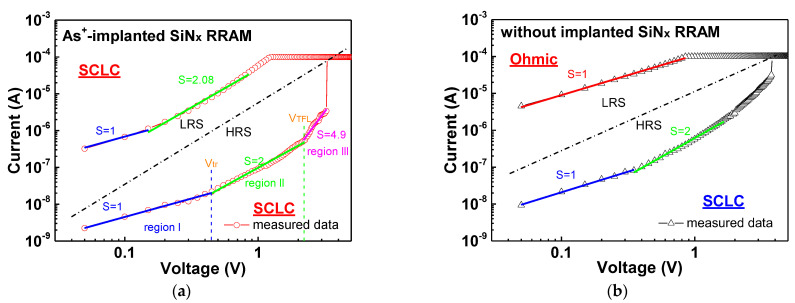
The analyzed *I–V* curves of the (**a**) As^+^-implanted and (**b**) nonimplanted SiN_x_ RRAM devices.

**Figure 6 nanomaterials-11-01401-f006:**
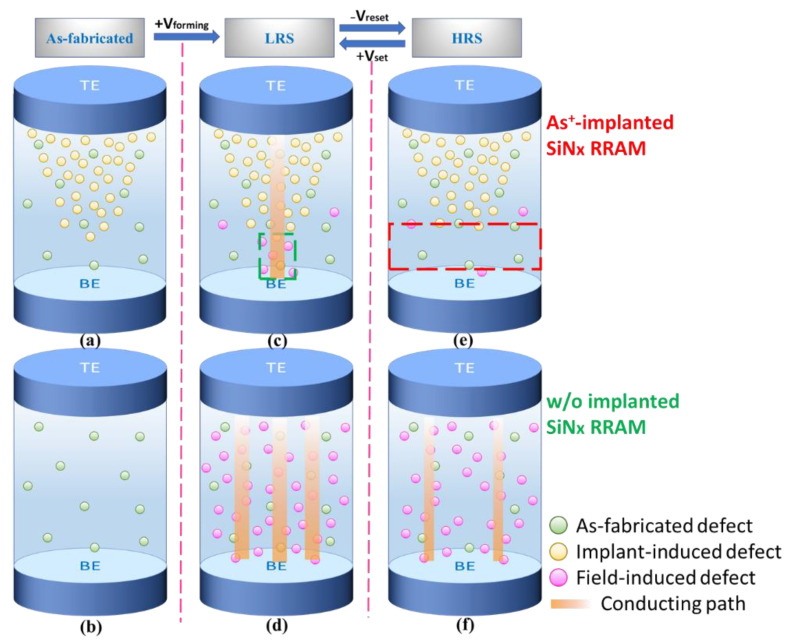
The schematic diagram of defect distribution and potential resistance switching characteristics in As^+^-implanted and nonimplanted SiN_x_ RRAM devices. (**a**), (**c**) and (**e**) are the As-fabricated state, LRS and HRS of the As^+^-implanted SiN_x_ RRAM. (**b**), (**d**) and (**f**) are the As-fabricated state, LRS and HRS of the nonimplanted SiN_x_ RRAM. Thinner conducting path lines in (**f**) because it is only leakage current.

**Table 1 nanomaterials-11-01401-t001:** The operation distribution performances of various SiN_x_ RRAM devices.

Reference	Switching Layer Materials	Thickness (nm)	CVs of V_set_ and V_reset_ (D2D)	CVs of V_set_ and V_reset_ (C2C)
15	PECVD-SiN_x_	25	18.3%/23.2%	14%/21.4%
15	PVD-SiN_x_	25	10.7%/12.1%	11.3%/11.4%
16	PECVD-SiN_x_	7.5	29%/17.77%	--
16	LPCVD-SiN_x_	7.5	16%/7.59%	--
This work	As^+^-implanted PECVD-SiN_x_	35	10.7%/9.8%	2.2%/3.8%

## Data Availability

The data presented in this study are available on request from the corresponding author. The data are not publicly available due to privacy.

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
