# Peer review of "Improved Device Distribution in High-Performance SiNx Resistive Random Access Memory via Arsenic Ion Implantation"

_nanomaterials, 2021, doi:10.3390/nano11061401_

Round 1
Reviewer 1 Report
The paper demonstrates that by applying As+ implantation to SiNx-based RRAM devices, the resisitive switching behavior significantly improves. This is a very interesting and useful result.
The authors give an explanation of the effect of implantation from a microscopic approach that involves the distribution of both defects induced by the implantation process itself and those field-induced. Some issues should be explained in a little more depth in order to improve the readability of the paper:
1- In Fig. 6.a, the high concentration of defects generated by the implantation could have increased the conduction of the SiNx layer in its upper area, so that the effective insulator layer was reduced to the area near the bottom electrode. Have any resistivity measurements been carried out that can shed light on this question?
2- What nature could be attributted to the conducting filament in Fig. 6.c?
In Fig. 6.f it is indicated that for the non-implanted sample there is only leakage current. If no conductive filaments are formed/dissolved, how is the shape of the I-V curves at set/reset transitions in Fig.1.b justified?

Reviewer 2 Report
This work presents the electrical characteristics of SiNx-based resistive switching MIS structures and a comparative study between SiNx layers with and without an As ion implantation. This is a subject of interest in the quest for highly reliable RRAM devices with low cycle-to-cycle and device-to-device variability. The present work proposes the use of an ion implantation step to create extensive structural damage in the dielectric material in order to reduce the involved voltages and currents in the device operation. Although this is an interesting approach, there are several aspects in the conducted research, the methodology used and the discussion of the obtained results that should be clarified and addressed, in order to accept this manuscript for publication.
The following items need to be clarified:
1) A comparison between as-deposited PECVD SiNx and As- ion implanted layers is first made in terms of forming voltage (Fig. 1). Since the forming voltage is, in fact, the dielectric breakdown voltage of the layers under study, to compare the dielectric strength of two materials, it is compulsory to make a statistical analysis of the obtained results in a significant number of samples. The results in 2 devices cannot be used to draw a conclusion. If this statistical analysis is done, please report the obtained results.
2) According to what is shown in Fig.1, the forming process has been different depending on the type of dielectric material of the MIS structure, using a higher current compliance limit in the case of as-deposited layers than in the case of ion-implanted layers. Having used different forming conditions, is the comparison of the resistive switching performance of the two types of devices significant? Why not using the same current compliance in both cases?
3) In the retention experiments, a detailed description of them should be given. Where the devices put in the HRS by means of a voltage ramp and then the current at the read voltage was recorded? Was the device continuously biased at the read voltage and the current only recorded after 1, 10, 100, 1000 and 10000 s ? Was then the same methodology used for devices put in the LRS?
4) With respect to the endurance experiments, in the inset of Fig. 2(b), the pulse characteristics (amplitude and width) are given. Why are the pulses amplitude of +/- 6 V? The currents at 0.5 V in Fig. 2(b), were measured with a pulse of 0.5 V /2 us after applying the 6 V pulse? Please indicate the measurement procedure.
5) The results shown in Fig. 3(b), correspond to the distribution of the set and reset voltages for 50 RS cycles. Were the cycles performed using a voltage ramp between the same voltage limits? Please clarify.
6) In terms of cycle-to-cycle and device-to-device variabilities, the reported results correspond only to the set and reset voltages. What about current at the read voltage in both resistance states? These are very relevant in the RRAM operation. Please report the obtained results to clearly show the better performance of the ion-implanted samples.
7) In the device-to-device variability results presented in Fig. 3(a), are the results of the 25 devices measured, the mean value of 50 RS cycles? Please, clarify.
8) In the manuscript, to assess the predominant electrical conduction mechanism in each resistance state, only the results in one voltage polarity are reported. In both types of SiNx layers, it is concluded that the conduction is either ohmic or space charge limited. Both are bulk-limited mechanisms, i.e., they are not electrode-limited. As a consequence, the same conduction mechanisms should be predominant in the case of negative polarity. Which are the obtained results in this polarity? In fact, according to the results shown in Fig. 1(b), there is a clear difference between negative and positive voltages.
9) What are the electrical conduction mechanisms before the forming process?
10) In Fig. 4, the results of the simulation of the As ion implantation are given, but no indication of the values of the Y axis are given. Please, comment on this.
In addition to these items to be clarified, I would like to make the following comments:
1) In the title of this work, the term high performance is used to refer to the SiNx based RRAM used. The reported characteristics of these devices are not as prominent as to deserve the “high performance” adjective. In the literature, there are several other types of RRAM devices which have a much higher performance that the devices in this work (for example, HfO2-based structures). The “Improved Device Distribution” in the title is misleading. In the manuscript, it is claimed that the device-to-device variability is improved, or that the set and reset voltages distributions are tighter in the case of As-implanted layers. In view of the above comments, please consider another title to correctly reflect the contents of the paper.
2) The number of self-citations is very high and most of them are not necessary as they do not support a statement in the manuscript. In the introduction, 42 references are cited related to RRAM devices, although the number of references associated to the different types of dielectric materials that are being investigated and with better performance, is not balanced. A newcomer in the field would think that GeOx, for instance, has deserved the same interest as HfOx, which is not the case. Please, reconsider self-citations.
3) Being the subject of this work, the electrical characteristics of SiNx, previous works have to be mentioned already in the introduction, such as ref. [35]. In addition, discussion about the interest of SiNx for RRAM would be appreciated.
4) In the introduction, the use of As as the ion implantation species is substantiated with the statement: “Besides, As is a semimetal; however, its property becomes a semiconductor with a bandgap of 1.2–1.4 eV if amorphized [43], which is used for ion implantation in this case.” In an ion-implantation process, the As+ ion is implanted into the material and these ions act as a dopant, but an As layer is not created. So, this statement has nothing to do with the role played by As in the SiNx layer. Please, consider removing this statement.
5) Please carefully review the English language. For example, dose is the correct term for the number of ions/cm2 in an ion implantation process, not dosage, or Poole-Frenkel instead of poole-frenkel,..
6) According to the discussion in this work, the resistive switching behaviour is attributed to the traps existing in the SiNx layer, either as-deposited or with As-implanted. Is the Ni electrode not playing a role? Some references in the literature point out the role of Ni, such as:
- Lu et al., “Investigations of conduction mechanisms of the self-rectifying n+Si-HfO2-Ni RRAM devices,” IEEE Trans. Electron Devices, vol. 61, no. 7, pp. 2294–2301, 2014.
- Rodriguez-Fernandez et al., Resistive Switching with Self-Rectifying Tunability and Influence of the Oxide Layer Thickness in Ni/HfO2/n+-Si RRAM Devices. IEEE Trans. Electron Devices, vol. 64, No. 8, pp. 3159-3166, 2017.
7) Is it necessary to cite references related to resistive switching devices when referring to the different conduction mechanisms in thin dielectric layers? Please, reconsider the references used. Ref. 45 is a review that can be referenced to cover all the mechanisms. These are conduction mechanisms that are well established and that are reported in general books.
8) According to the proposed picture to explain the observed behaviour, will a SiNx layer as thin as the layer of the implanted sample without a significant amount of defects (red box in Fig. 6(c)) behave as the 35 nm implanted SiNx layer? Will the use of another ion, instead of As, have the same effect?
Reviewer 3 Report
- In Fig. 1, after As doping, the HRS and LRS currents decreased which means both HRS and LRS became more resistive. It is counter-intuitive that As doping of SiN will increase the overall resistance. In Fig. 6, shows a schematic explaining this phenomenon. In Fig. 6, the As ions in a gradient distribution is understandable. In Fig. 4 they show a mask to limit the As ions doped area. Please elaborate the size of the mask opening and the reason for ion implantation doping concentration and energy. Is there any correlation between ion implantation doping concentration and/or energy and device performance?
- Please show the actual device microscope picture. What size is the device?
- Also SEM side image of the SiNx device.
- The fact that a specific memory window ( 1.73×103 resistance window) is mentioned probably shows that the paper has data from only one or two devices. Please have a figure that shows the device to device variations (i.e. measurement results from several devices and the comparison with doped and undoped)
- The forming and SET, RESET voltages are quite high. The thickness of SiNx was 35nm. Any reason you didn’t lower the thickness to have lower operating voltages? Please explain in the main text.
Round 2
Reviewer 2 Report
It is the opinion of this reviewer that the role of a reviewer is to make suggestions and comments to a manuscript, trying not to be disrespectful and not to make impolite remarks, and one would expect the same from the authors.
Instead, the authors' response to questions 11 and 12, using red fonts and capital letters is inappropriate.
This is rather disappointing.
Reviewer 3 Report
A device paper without the actual device microscopic image or electron microscope image is not acceptable especially if it is an experimental paper.
It is also very important for the readers to understand the structure and its characteristics in a visualized way.
Without such images in the main text, I cannot accept this paper for publication. Please include at least one microscopic image showing the device size and one electron microscope image (SEM or TEM or any electron microscopy) in the main figure.
